# PRETEXT TASKS SELECTION FOR MULTITASK SELF-SUPERVISED SPEECH REPRESENTATION LEARNING

## ABSTRACT

Through solving pretext tasks, self-supervised learning leverages unlabeled data to extract useful latent representations replacing traditional input features in the downstream task. In audio/speech signal processing, a wide range of features where engineered through decades of research efforts. As it turns out, learning to predict such features (a.k.a pseudo-labels) has proven to be a particularly relevant pretext task, leading to useful self-supervised representations which prove to be effective for downstream tasks. However, methods and common practices for combining such pretext tasks for better performance on the downstream task have not been explored and understood properly. In fact, the process relies almost exclusively on a computationally heavy experimental procedure, which becomes intractable with the increase of the number of pretext tasks. This paper introduces a method to select a group of pretext tasks among a set of candidates. The method we propose estimates calibrated weights for the partial losses corresponding to the considered pretext tasks during the self-supervised training process. The experiments conducted on automatic speech recognition, speaker and emotion recognition validate our approach, as the groups selected and weighted with our method perform better than classic baselines, thus facilitating the selection and combination of relevant pseudo-labels for self-supervised representation learning.

## 1 INTRODUCTION

Self-supervised learning (SSL) methods usually rely on a supervision obtained from the data itself through solving specific pretext tasks leveraging the underlying structure of the considered data (Doersch et al., 2016; Arandjelovic & Zisserman, 2018). This technique is used in various domains including image processing (Misra & Maaten, 2020; Jing & Tian, 2020; Grill et al., 2020), natural language understanding (Chen et al., 2020b; Du et al., 2020; Lan et al., 2019) or speech and audio processing (Baevski et al., 2020b; Liu et al., 2020; Jiang et al., 2020). It offers numerous advantages, such as the independence from labeled data, stronger performance on downstream tasks, more robust models and an easier transfer to low-resource setups (*e.g.*, low-resource languages) (Baevski et al., 2020b; Jing & Tian, 2020).

The numerous existing SSL approaches are characterized by the nature of the pretext tasks they solve. For instance, common techniques include predictive coding (Baevski et al., 2020b; Liu et al., 2020; Song et al., 2020; Zhang et al., 2020; Hsu et al., 2021), pseudo-label learning (Pascual et al., 2019; Ravanelli et al., 2020), auto-encoding (Renshaw et al., 2015; Algayres et al., 2020), triplet-loss learning (Shor et al., 2020; Peplinski et al., 2020), generative modelling (Khurana et al., 2020) or contrastive learning (Saeed et al., 2020; Jiang et al., 2020). More precisely, these pretext tasks may be defined through the choice of pretext labels, hereafter referred to as *pseudo-labels*. The automatic extraction of pseudo-labels for SSL (*i.e.* from the data itself) is common in many application domains, such as computer vision (Noroozi & Favaro, 2017; Gidaris et al., 2018), music processing (Hung et al., 2019; Wu et al., 2021) and speech processing (Pascual et al., 2019; Shukla et al., 2020), and is commonly referred to as *multitask self supervised learning*. In the specific context of speech processing, the process of designing pseudo-labels may benefit from decades of research in signal processing. For instance, potential candidates are pitch estimators, energy-based features, voicing state and many more.

As demonstrated by Pascual et al. (2019), multitask speech representation learning is a powerful tool to build representations that are beneficial for a wide range of distinct downstream tasks, by combining different pseudo-labels which "intuitively" correspond to these tasks. Unfortunately, there is no clear understanding of how these pseudo-labels may interact when optimised together, and therefore, no common practice of how to select groups of pseudo-labels to obtain better performance on a known downstream task. As a matter of fact, this design process has been essentially driven by empirical validation and there is therefore no evidence that the obtained model is even the best one. This empirical approach can rapidly become intractable with modern SSL architectures which may contain hundreds of millions or billions of parameters trained on thousands of hours of speech, not to mention the carbon footprint of such pseudo-label searches. For instance, the self-supervised training of a single state-of-the-art large wav2vec 2.0 model (Baevski et al., 2020b) on $53.2k$ hours of speech requires $128$ GPUs for $5.2$ days.

This work aims at providing a clear, efficient and theoretically motivated procedure for pseudo-label group selection and weighting based on conditional independence (CI). The method presented allows one to design ahead of training the most adapted multitask self-supervised speech representation learning model which perfectly suits the considered downstream tasks. Such an approach may also enable researchers to save a substantial amount of time and computation usually devoted to pseudo-label search. Hence, the contributions of this work are fourfold:

1. Introduce a theoretically motivated and computationally efficient method for the selection of pseudo-label *groups* among a set of candidates and with respect to the considered downstream tasks (Sections 3 and 4).

2. Validate empirically the proposed approach with a first model SSL model relying on different sets of pseudo-labels corresponding to the ones obtained for three considered speech tasks. (Sections 5).

3. Extend our method to the SOTA wav2vec 2.0 to enhance its performance (Section 6).

4. Release the code base developed with SpeechBrain (Ravanelli et al., 2021) for replication and to encourage further investigations.[1]

The conducted experiments demonstrate that the proposed method allows for a more intelligent, *i.e.* better informed, pseudo-label group selection for multitask SSL settings. Indeed, we find that the models built with the proposed method obtain a word error rate and an equal error rate, respectively, $31.6\%$ and $27.4\%$ lower than the baseline, without the need for any empirical search.

## 2 RELATED WORKS AND MOTIVATIONS

SSL recently became a key component to achieve good performance on downstream tasks especially with low-resource setups, either in speech (Baevski et al., 2020b; Conneau et al., 2020), natural language processing (Lan et al., 2019; Chen et al., 2020b) or computer vision (Gidaris et al., 2019; Misra & Maaten, 2020; Jing & Tian, 2020). Due to its very nature, SSL relies on large amounts of unlabeled data used to train large deep neural networks over long periods of time. It it thus crucial to understand properly what makes a good SSL model to lower the amount of computation and time needed to obtain the best downstream performance.

**SSL for Speech.** Self-supervised learning for speech has recently enabled researchers to reach state-of-the-art results on various speech processing tasks (Fan et al., 2021). The most successful models rely on predictive and contrastive objectives (Baevski et al., 2020b; Chung et al., 2019; Hsu et al., 2021; Shor et al., 2021) performing well across the different tasks even in low-resource settings. This led to the design of different benchmarks evaluating the self-supervised representations in different languages (Yang et al., 2021; Evain et al., 2021). However, in contrast to this proposition, these works have not tried to motivate beforehand the different choices made in the self-supervision pipeline.

**Understanding SSL.** A few works have tried to shed some theoretical light on the mainly empirical field of self-supervised learning. Following the different paradigms in SSL, various tracks have

---

[1] https://github.com/iclrsubmitter22/iclr_2022_submission

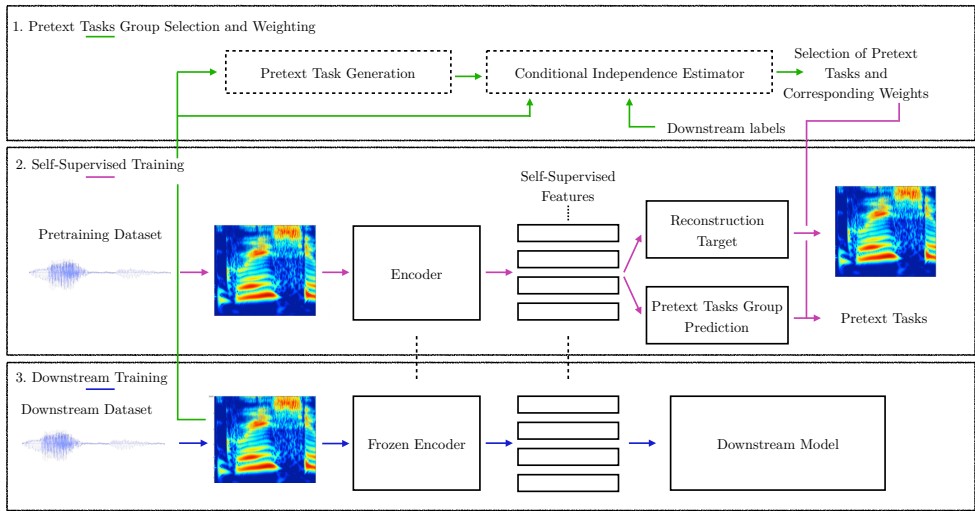

Figure 1: Illustration of the training pipeline. The three steps are depicted: 1. Selecting the group of pseudo-labels and their corresponding weights; 2. SSL training with the selected pretext task; 3. Training on the downstream task with the pretrained SSL model.

been followed to understand what makes for a good self-supervised representation, exploring different approaches (Lee et al., 2020; Arora et al., 2019; Wei et al., 2020). On the one hand, contrastive learning (Oord et al., 2018; Chen et al., 2020a) has been advocated both theoretically and empirically to achieve a balance in the mutual information between alternative representations of the data, keeping just enough shared information to keep the class-related content (Tschannen et al., 2020; Tian et al., 2020; Bachman et al., 2019). In a recent work from Li et al. (2021), independence testing has been used to produce better transformations in contrastive learning settings for image representations. Predictive learning, on the other hand, requires the model to predict masked elements in sequential data. This technique is powerful on downstream tasks that can be reduced to a masking problem, as suggested by research on language modeling (Saunshi et al., 2020). However, all these works have been focusing on computer vision or text-related applications, and none of them addressed the multi-tasked self supervision problem.

**Multi-task self-supervised learning.** While the literature on multi-tasking in self-supervised learning remains scarce, it has been shown in classic supervised learning settings, that through estimates of similarity between tasks or thorough empirical testing, several tasks can take advantage of being solved with a common encoder (Zamir et al., 2018; Dwivedi & Roig, 2019; Shafey et al., 2019; Chen et al., 2015). More specifically, combining pretext tasks with SSL has been mainly explored in computer vision and speech (Pascual et al., 2019; Ravanelli et al., 2020). Pretext tasks such as Jigsaw (Doersch et al., 2016), colourisation and rotation (Gidaris et al., 2018) have been combined successfully to improve downstream performance (Kim et al., 2018; Shin'ya Yamaguchi et al.). The two closest works to our line of research are from Lee et al. (2020) and Doersch et al. (2017). The former shows that a theoretical link can be established between conditional independence and an improvement of the performance on the downstream task, while the latter proposes to select layers from a multitask self-supervised encoder according to the pretext task to be solved. However, in both cases, the studies do not offer practical and theoretical solutions to select groups of pseudo-labels to build an adapted SSL model that will perform well on the considered downstream tasks.

**Group feature selection.** Finally, feature selection, and especially group feature selection is another close and inspiring field given the problem we consider. The relationship and interactions between features have been largely investigated in the supervised learning literature (Guyon & Elisseeff, 2003). This led to multiple solutions to the feature group selection problem, including LASSO-based techniques (Yuan & Lin, 2006), or multiple kernel formulations (Sonnenburg et al., 2006; Rakotomamonjy et al., 2007). However, these works do not involve any self-supervision, and links

between feature selection and self-supervision design and pretext-task selection are yet to be proved. We will further consider these lines of works for concurrent baselines.

With this work, we aim at shortening the process of designing SSL models while giving insights on the pseudo-label importance and the underlying mechanisms between pretext and downstream tasks at the same time. We decided to experiment with speech due to the lack of literature on this domain for multitask SSL, and for the various pseudo-labels available, which are based on decades of signal processing research. The whole pipeline starting from the acoustic feature extraction to the downstream task scoring follows three major steps summarized in Figure 1. First, for every downstream task, our method produces a pretext task selection and weighting. Then, a SSL model is trained, before being used as a feature extractor front-end to one or many downstream tasks.

## 3 Conditional independence for utility estimation

As a first step, we require a function that estimates the utility of learning to solve a pretext-task to improve the performance on the downstream task. We use an estimation of the conditional independence between the pseudo-label values and the downstream data points given the downstream labels. Hereafter, we explain the theoretical motivations and describe the computation steps.

### 3.1 Problem definition and intuition

Let $X$, $Y$ and $Z$ be, respectively, the downstream data points, their downstream labels and their pseudo-labels. Let also $\mathcal{C}$ be the set of possible downstream classes. As an example, if one considers speaker recognition as a downstream task, $X$ would be the speech samples, $Y$ the speaker IDs, $\mathcal{C}$ the set of unique speaker IDs, and $Z$ a computed signal feature, such as the fundamental frequency.

As stated in Section 2, Lee et al. (2020) linked the utility of a pseudo-label ($Z$) to the conditional independence (CI) between $Z$ and $X$ given $Y$. The approach prescribes that, given the labels $Y$, one may seek to *quantify how much it is possible to predict the pseudo-labels $Z$ without knowing much about $X$*. The authors bounded, under certain assumptions, the downstream classifier error with a function of the downstream training set size, and a measure of the CI. More precisely, the main theorem shows that the bounding function decreases linearly with the downstream-task dataset size ($M$) and quadratically with the CI, thus making it a potential estimator for pseudo-label utility.

The proposed function depends on the final downstream task to be solved. This is motivated by two main reasons. First, it can be seen through the large literature on feature selection for various speech or computer vision tasks (Liu et al., 2020; Serizel et al., 2017; Schuller et al., 2007; Wang et al., 2019), that different tasks require the description of different aspects of the data. This suggests that different downstream tasks may perform better after different pre-trainings. A second argument is the difficulty to evaluate representations quality intrinsically, *i.e.* independently from the choice of a particular downstream task. A few metrics and tests (Schatz et al., 2013; Carlin et al., 2011; Lakhotia et al., 2021) have been proposed for speech, but the correlation between these and downstream-task performance has not been clearly identified (Algayres et al., 2020; Gump et al., 2020).

The principal issue with CI is the difficulty of computing good estimates of how much two variables are independent given a third one on realistic data (Shah & Peters, 2018). In a previous work (not cited to respect the double blind reviewing), we proposed a simple way to get an estimation of the conditional independence. This method has proven effective for individual pretext task selection, as the utility estimator correlated highly with the final downstream performances. In our case, the considered pseudo-labels are not independent of the speech samples, as they are signal features. The approach resorts to performing classic independence testing on data sliced by the downstream classes, to check whether this dependence remains given the downstream labels information.

### 3.2 Conditional independence estimator computation

This section details the computation of the conditional independence estimate that is used as a measure of pseudo-label utility. Let $X = \{x_i\}_{i \in \{0,...,M\}}$ with $M$ being the cardinal of $X$ and $x_i$ data samples (*e.g.* Mel-band spectrogram for audio and speech). Every sample $x_i$ has a corresponding downstream label $y_i$ and an automatically generated pseudo-label $z_i$. We assume that $y_i$ is discrete reducing the task to a classification problem such as with speaker ID for speaker recognition. We

also assume that for every pretext-task $Z$, a single $z_i$ value corresponds to each $x_i$. In our case, $z_i$ values are the mean of the frame-wise pseudo-label values.

For independence testing, we decided to rely on the kernel-based Hilbert Schmidt Independence Criterion (HSIC) (Gretton et al., 2007) for two reasons. First, HSIC has already proven successful for textual data in testing statistical dependence between translated sentences (Gretton et al., 2007). Second, kernel-based techniques facilitate the handling of multivariate and varying-length data such as speech, as the estimation then boils down to the computation of a similarity measure between the considered variables.

**Computation steps.** The estimation of the CI of a pseudo-label $Z$ for a downstream task $(X, Y)$ consists of three steps. We start by splitting the data samples $X$ according to the downstream (discrete) classes. Then, we compute for every downstream class $c \in \mathcal{C}$, the kernel matrices $K_c$ and $L_c$ representing the similarity measures for the data samples, and the pseudo-labels, respectively. Finally, we perform the independence test for every split group using $K_c$ and $L_c$ and aggregate the estimates with a weighted mean taking into account the number of samples per downstream class. Thus, for two speech samples $x_i$ and $x_j$, holding two pseudo-label values $z_i$ and $z_j$, the coefficients of the similarity matrices $K_c$ and $L_c$ are computed as follows:

$$K_{ij} = K(x_i, x_j) = \cos(GD(x_i), GD(x_j)), \quad L_{ij} = RBF(z_i, z_j), \tag{1}$$

with $GD(.)$ the Gaussian Downsampling function (more details in the appendix A.7) , $\cos(.,.)$ the cosine similarity, and $RBF(.,.)$ the Radial Basis Function kernel, defined as:

$$\cos(x, x') = \frac{trace(x^T x')}{||x||.||x'||}, \quad RBF(z, z') = \exp(-\frac{||z - z'||^2}{2\sigma^2}), \tag{2}$$

where $\sigma$ is the width of the RBF kernel and $trace(.)$ the sum of elements of the main diagonal. Note that we compute the matrices $K_c$ and $L_c$, for each group of samples sharing the same downstream class $c \in C$. Hence, $K_c$ and $L_c$ correspond to the definitions above, but restricted to the points with $c$ as a downstream label. For each downstream class $c$, and as in Gretton et al. (2007), the HSIC value is given by:

$$HSIC_c(X, Z) = \frac{1}{n_c^2} trace(K_c H_c L_c H_c), \tag{3}$$

with $H_c = I_{n_c} - \frac{1}{n_c} 1_{n_c} 1_{n_c}^T$, $n_c$ being the number of points with downstream label $c$, and $1_{n_c}$ a vector of ones of size $n_c \times 1$.

The HSIC value is non-negative and corresponds to the Hilbert norm of their cross-covariance matrix. It is used to characterize the independence of the two considered quantities. Intuitively, the HSIC value is high if samples similar in $K_c$ are similar in $L_c$. Therefore, the lower this value is, the more independent the two arguments of HSIC are and the better the pseudo-label should be for self-supervision before fine-tuning on the downstream class. The final value for a given pseudo-label and a downstream task is expressed as:

$$HSIC(X, Z|Y) = \frac{1}{M} \sum_{c \in \mathcal{C}} HSIC_c(X, Z) \times n_c. \tag{4}$$

**Computational efficiency.** Efficiency is one of the key motivations of this work, and the gain in time observed with our approach is considerable. The $K$ and $L$ matrices used for the CI estimate are only computed for the downstream datasets. But since these datasets may get very large, we can sample among the downstream classes to keep the computations tractable as shown in appendix A.3.

## 4 PRETEXT TASK GROUP SELECTION AND WEIGHTING

While we now are able to predict the utility of every considered pretext task independently, the next step remains to define a clever way to combine them optimally within the same pre-training process. Hence, we introduce a method to select a group of pseudo-labels and weight their respective losses to increase or decrease their importance in the self-supervised representation. Such an optimisation of the latent representation is expected to provide better downstream performance. Our method minimises the conditional dependence between the resulting group pretext task, entailing the prediction of a selected pseudo-label group and the downstream samples given the downstream labels.

Given a set of $k$ possible pseudo-labels $(Z_i)_{i \in [0,k]}$, we seek to estimate a set of parameters $(\lambda_i)_{i \in [0,k]}$ weighting the loss of every pseudo-label $k$ during the pre-training phase. Hence, we define a grouping pseudo-label $Z_\lambda$ as an orthogonal concatenation of $(Z_i)_{i \in [0,k]}$ weighted with $(\lambda_i)_{i \in [0,k]}$ : $Z_\lambda = (\lambda_1 Z_1, ..., \lambda_k Z_k)$.

The custom conditional HSIC computation pipeline described above is fully differentiable with respect to $(\lambda_i)_{i \in [0,k]}$ as proved in appendix A.1. In the HSIC computation, the data similarity matrices $\{K_c\}_{c \in C}$ are independent of $Z$ and therefore of $\lambda$. Only the pseudo-label similarity matrices $\{L_c\}_{c \in C}$ are changed. For every downstream class $c$, $L_c$ is defined as:

$$[L_c]_{i,j} = RBF((Z_\lambda)_i, (Z_\lambda)_j) = \exp(\frac{-1}{2\sigma^2} \sum_{h=1}^{k} \lambda_h ||z_{h,i} - z_{h,j}||_2^2), \tag{5}$$

with $z_{h,i}$ denotes the mean value of the $h$-th pseudo-label for the $i$-th data point in the dataset.

## 4.1 CONSTRAINTS ON THE WEIGHTS

The conditional-independence based utility estimator must be optimized with respect to the weighting parameters $(\lambda_i)_{i \in [0,k]}$ and three constraints.

First, the parameters $(\lambda_i)_{i \in [0,k]}$ must be positive, as they are used as weights for the corresponding losses. A negative weighting loss would lack interpretability as it could imply that the self-supervised model should "unlearn" the corresponding pretext task. This may be the case for adversarial learning methods, but we are not considering this case in the present work.

Second, the value of the weights must be high enough. Indeed, the presented method for estimating the conditional independence assumes that the considered pseudo-label $Z$ is not independent of $X$. It is fortunately true for speech features, as $Z$ is a function of $X$, but not always valid. For instance, with $(\lambda_i)_{i \in [0,k]} = 0$, the utility estimator would be null and thus the lowest (*i.e.* the best), but it would break the assumption of non independence between $Z$ and $X$. Furthermore, the $HSIC$ value decreases with positive decreasing values of $(\lambda_i)_{i \in [0,k]}$ and we thus need to ensure that the sum of the weights is significantly higher than zero, or it would mean that we are not really doing multi-task learning as the losses of the pseudo-labels would be barely considered.

Finally, for a fair comparison between the weighting choices during the optimization, the sum of the weights should remain constant. In the following, the sum of the $(\lambda_i)_{i \in [0,k]}$ is fixed to one and the problem is summarized as follows:

$$\min_{\lambda \in \mathbb{R}^k} \quad HSIC(Z_\lambda, X, Y), \text{ s.t. } Z_\lambda = (\lambda_1 Z_1, ..., \lambda_k Z_k), \ \lambda_i \geq 0, \ \forall \, i \in [0,k], \ \sum_i \lambda_i = 1. \tag{6}$$

To minimize the estimator quantity while respecting the constraints, the weights used in the computation of the CI value are the softmax output of free learnable parameters $(W_i)_{i \in [0,k]}$. Softmax enforces the conditions while being differentiable and the problem becomes:

$$\min_{W \in \mathbb{R}^k} \quad HSIC(Z_\lambda, X, Y), \text{ s.t. } \lambda = Softmax(W), \ Z_\lambda = (\lambda_1 Z_1, ..., \lambda_k Z_k). \tag{7}$$

## 4.2 WEIGHTS SPARSITY

Another trait that is desirable for the weighting vector is sparsity. If a few pseudo-labels are not needed for the given downstream task, they would rather be discarded than given a low weight. First, this would save computation time including the extraction of the pseudo-labels, and their extraction and prediction during the self-supervised training process. Second, it would help with transparency to understand what features are included or not in the latent space. This sparsity property is also related to feature selection such as with LASSO (Yuan & Lin, 2006). To ensure the sparsity of the output weighting vector, while maintaining the desired property of differentiability, we choose the sparsemax function (Martins & Astudillo, 2016) to replace the softmax in eq. 7.

## 5 EXPERIMENTAL STUDY

This section details the experiments validating the introduced estimator. It describes the selection and weighting processes (Section 5.1), the SSL models (Section 5.2), the downstream tasks (Section 5.3), and the obtained results (Section 5.4).

## 5.1 GROUP SELECTION AND WEIGHTING

Individual pseudo-labels of interest are obtained with the OpenSmile library (Eyben et al., 2010). We decided to focus on markers mostly related to prosody and spectral descriptors as the signal processing literature commonly associates them to the three considered downstream tasks (*i.e.* speech, speaker and emotion recognition). Selected pseudo-labels include: *Loudness, F0, Voicing, $\alpha$ Ratio* (Sundberg & Nordenberg, 2006), *Zero Crossing Rate, L1 Norm of Rasta Spectrum* (Hermansky et al., 1992), *log of Harmonicity to Noise Ratio* (Murphy & Akande, 2005). Then, and according to Figure 1 (*step 1*), we group these pseudo-labels based on their weights, *i.e.* by optimising equation 7 to obtain the different $\lambda$ values associated to each one of them.

Comparative baselines follow common feature group selection strategies or natural intuitions. The first one simply bundles all the pseudo-labels together without any weighting (*i.e.* $\lambda = 1$ for all pseudo-labels) as proposed for PASE (Pascual et al., 2019). As SSL group pretext-task selection is yet to be fully explored, the two other baselines come from the feature selection literature as it represents the closest field with well established techniques. The CI-based pseudo-label selection is thus compared to Recursive Feature Elimination (RFE, Guyon et al. (2002)) and Maximum Relevance Minimum Redundancy (MRMR, Peng et al. (2005)). We also add an experiment with the remaining pretext tasks after the MRMR selection, this is to show the effect of learning supposedly unrelated workers, we will call it "Remaining". More details about these baseline algorithms are given in Appendix A.9, while Appendix A.11 shows the workers selected and their corresponding weights for every experiment. Noise (HNR) seems to be the most important information to learn to predict for speaker recognition while fundamental frequency is privileged for ASR.

## 5.2 SELF-SUPERVISED TRAINING

In the second step of Figure 1, the SSL model learns to predict the selected pseudo-labels. For every one of those, the loss is multiplied by the corresponding assigned weight. Based on previous work conclusions (Ravanelli et al., 2020; Jiang et al., 2020) and apart from the considered pretext task, the network learns to reconstruct the input Mel spectrograms, and to compute 40-dimensional Mel-Frequency Cepstral Coefficients (MFCC) feature vectors. These targets are usually kept to avoid information loss harming heavily downstream performance and are used in all our experiments. For a given weighting vector $(\lambda_i)_{i \in [0,k]}$, the self-supervised loss is defined as:

$$L_{SSL} = MSE_{mel} + MSE_{mfcc} + \sum_{i=1}^{k} \lambda_i \ell_1(Z_i), \tag{8}$$

with $MSE$ the classic mean squared error computed for Mel spectra ($MSE_{mel}$) and MFCC ($MSE_{mfcc}$), and $\ell_1(Z)$ the $\ell_1$-loss of the pretext task related to pseudo-label $Z$.

Prior to extending our method to state-of-the-art architectures such as wav2vec 2.0 that are particularly costly to train, we propose to first employ a PASE-like model to empirically validate the approach. Hence, the encoder is composed of three distinct parts: a VGG-like feature extractor (Simonyan & Zisserman, 2015), a bidirectional LSTM, and a two-layered dense neural network. All the details of the architecture are given in the appendix A.5. Then, and inspired by Ravanelli et al. (2020), the encoder is followed by simple one-layered predictors voluntarily limited in capacity.

**SSL dataset.** The SSL model is optimised on the training set of the English Common Voice dataset (version 5.1, 700 hours of training, Ardila et al. (2020)). Common Voice is a collection of speech utterances from worldwide users recording themselves from their own devices. Hence, the closeness to natural settings makes it a suitable choice for self-supervised learning. 700 hours of speech is a relatively small amount compared to what is generally used for state-of-the-art SSL models. However, we believe it is a sound choice as this is generally greater than what is typically available in SSL use-cases like low-resource languages. We decided to not use the LibriSpeech dataset for pre-training as it is part of our downstream evaluation protocol hence alleviating a strong bias.

## 5.3 DOWNSTREAM TASKS

Our proposed pseudo-label selection strategy is compared with the two baselines on three different downstream tasks leading to different groups of pseudo-labels: automatic speech recognition (ASR, with LibriSpeech 100 hours) speaker recognition (SR, with VoxCeleb 1), and emotion recognition (ER with IEMOCAP). Datasets and downstream architectures are inspired from the SUPERB

Table 1: Results observed with the proposed selection strategies on the two considered downstream tasks. Word Error Rate (WER) Equal Error Rate (EER), and Accuracy (Acc) are expressed in percentage and used for LibriSpeech 100 hours, VoxCeleb1 and IEMOCAP respectively (*i.e.* lower is better). ASR results are given with and without Language Modeling (LM). All SSL models contain $16.3M$ neural parameters.

| Models | LibriSpeech *(WER % ↓)* | | VoxCeleb1 *(EER % ↓)* | IEMOCAP *(Acc % ↑)* |
|---|---|---|---|---|
| | *No LM* | *LM* | | |
| PASE+ (Ravanelli et al., 2020) | 25.11 | 16.62 | 11.61 | 57.86 |
| vq-wav2vec (Baevski et al., 2020a) | 17.71 | 12.80 | 10.38 | 58.24 |
| Selections | | | | |
| All | $21.98 \pm 0.36$ | $11.70 \pm 0.27$ | $11.90 \pm 0.32$ | $56.4 \pm 1.3$ |
| MRMR | $18.94 \pm 0.34$ | $10.36 \pm 0.26$ | $10.56 \pm 0.31$ | $59.6 \pm 1.29$ |
| Remaining | $19.81 \pm 0.34$ | $11.65 \pm 0.27$ | $11.67 \pm 0.32$ | $58.8 \pm 1.29$ |
| RFE | $20.02 \pm 0.34$ | $11.42 \pm 0.27$ | $11.91 \pm 0.33$ | $55.8 \pm 1.3$ |
| Softmax | $\mathbf{13.17 \pm 0.28}$ | $\mathbf{8.00 \pm 0.23}$ | $9.24 \pm 0.29$ | $60.6 \pm 1.27$ |
| Sparsemax | $17.18 \pm 0.32$ | $10.41 \pm 0.26$ | $\mathbf{8.63 \pm 0.27}$ | $\mathbf{60.8 \pm 1.28}$ |

benchmark (Yang et al., 2021) for self-supervised learning representations and carefully described in Appendix A.5.3. Prior to downstream training, the SSL model are frozen to be used as a feature extractor with the new pipeline that is task-dependent. We do not use any data augmentation for a pristine comparison of the learned models.

## 5.4 RESULTS

Baselines detailed in Section 4 are respectively referred to as "*All*", "*RFE*" and "*MRMR*". All the details about the selection and weights are available in Appendix A.11. First, it is clear from the results reported in Table 1 that, for the considered downstream tasks, the two introduced strategies (*Sparsemax* and *Softmax*) perform better than the group selection baselines with a gain of 3.28 of EER for *Sparsemax* against the *RFE* approach on VoxCeleb, and 8.81 of WER with *Softmax* compared to the *All* baseline. Interestingly, simply bundling all the pseudo-labels together may lead to poor performance as observed on LibriSpeech with a very high 21.98% of WER obtained. Hence, *intuitively* building sets of labels could be harmful for the final representation. This motivates the need for a better pseudo-label selection strategy such as the one introduced in this work, as the WER dropped to 13.17%. As a comparison, the exact same architecture trained with Mel spectra only (*i.e.* no SSL) obtains a WER of 17.3% without LM. Hence, our method even further decreases the WER while being only pretrained with a reasonable amount of data (*i.e.* only 700 hours compared to a few thousands for common SSL techniques (Baevski et al., 2020b)). As expected, introducing the joint decoding with a language model strongly decreases the WER but also introduces a bias in our comparison as probabilities are smoothed with a third-party neural model. Nevertheless, and even in this scenario, our weighting strategy outperforms all the baselines. In the context of speaker recognition, *Sparsemax* beats *Softmax* with an EER 0.61 lower. For IEMOCAP, *Softmax* and *Sparsemax* weighting still perform the best among all methods. To investigate how strongly improvements are correlated to the task, we took the best learned model for LibriSpeech (i.e. softmax weighting) and fine-tuned it on VoxCeleb1 and IEMOCAP. It reaches an EER of 10.55% and an accuracy of 59.9% respectively. While it performs better than the baselines, the difference between these results and the best performing selections shows that the weightings are indeed task-related.

## 6 EXTENDING WAV2VEC 2.0 TO MULTITASK SSL

To the best of our knowledge, multi-task speech representation learning has not been scaled to a point where it could represent a state-of-the-art alternative. Contrastive predictive coding (Oord et al., 2018) based techniques like wav2vec 2.0 (Baevski et al., 2020b), on the other hand, currently trust most of the leader-boards for speech-related tasks. Recently, Sadhu et al. (2021) showed that combining a consistency loss and contrastive predictive coding improves the results of the wav2vec 2.0 architectures in noisy conditions. Following this idea, we propose to further validate our selection method with an extension of wav2vec 2.0 to multitask SSL to demonstrate its scaling capabilities.

Table 2: Results observed retraining the Wav2vec2 model with and without weighted pretext tasks using the sparsemax method. "Fr." and "Fine." also respectively refer to Frozen and Finetuned settings. Adding selected pretext tasks improves the donwstream performance on all three considered tasks. All models contain $100M$ neural parameters.

| Selections | LibriSpeech (WER % ↓) | | VoxCeleb1 (EER % ↓) | | IEMOCAP (Acc % ↑) | |
|---|---|---|---|---|---|---|
| | Fr. | Fine. | Fr. | Fine. | Fr. | Fine. |
| wav2vec 2.0 *BASE* | $17.93 \pm 0.33$ | $10.21 \pm 0.25$ | $7.20 \pm 0.26$ | $5.35 \pm 0.22$ | $56.6 \pm 1.2$ | $\mathbf{74.0 \pm 1.16}$ |
| wav2vec 2.0 *BASE* + Naive selection | $17.23 \pm 0.32$ | $10.10 \pm 0.25$ | $6.80 \pm 0.25$ | $\mathbf{5.05 \pm 0.21}$ | $57.4 \pm 1.3$ | $73.7 \pm 1.16$ |
| wav2vec 2.0 *BASE* -Sparsemax | $\mathbf{16.70 \pm 0.31}$ | $\mathbf{9.18 \pm 0.24}$ | $\mathbf{6.57 \pm 0.25}$ | $5.30 \pm 0.22$ | $\mathbf{59.5 \pm 1.29}$ | $\mathbf{74.0 \pm 1.16}$ |

Hence, the training loss is extended to:

$$L_{SSL} = L_{W2V} + \sum_{i=1}^{k} \lambda_i \ell_1(Z_i). \qquad (9)$$

We use the standard *BASE* wav2vec 2.0 first described in (Baevski et al., 2020b) as a SSL model and train it with the same Common Voice dataset. The pre-training pipeline is implemented within SpeechBrain. The trained *BASE* model has been compared to one obtained with the official Fairseq implementation from Baevski et al. (2020b), and results are strictly equivalent. The entire recipe alongside with the large set of hyperparameters needed to properly train a wav2vec 2.0 model are released under our anonymous repository and will be made available with SpeechBrain afterwards.

We follow the SUPERB benchmark conventions (Yang et al., 2021) both at the data and downstream architecture levels. Hence, and conversely to the previous experiments, the ASR system solely optimises the CTC criterion over characters. For each of the three tasks (*i.e.* ASR, SV, ER) we compare the standard *BASE* wav2vec 2.0 model with one trained following the sparsemax selection of multitask SSL. Sparsemax is chosen over softmax because it enforces the sparsity criterion and removes completely a few pseudo-labels from the training, which is one of the objectives of this work. Another experiment is led with a "naive" pretext-task selection where a constant weight of 0.5 is used across all signal-based pretext-tasks. Like the other experiments, the exact weights of each pseudo-label are reported in Appendix A.11. Each wav2vec 2.0 model required 24 NVIDIA Tesla V100 GPUs to train for 150 epochs (40 hours). Finally, we also propose to compare frozen and unfrozen (*i.e.* where the wav2vec 2.0 encoder is fine-tuned with the downstream task) SSL models.

It is clear from the results reported in Table 2 that our approach improves the performance over the standard wav2vec 2.0 framework for every considered downstream task. While adding pretext tasks naively improves the final performance, the difference in performance between the naive selection and the sparsemax weighting shows the benefit of our method in getting the best downstream performance. Unsurprisingly this difference is small (though statistically significant in all but one case), as the wav2vec 2.0 BASE is already powerful and the additional workers are anyway useful. Here, it is worth noting that the difference in performance compared to the literature mostly comes from the pre-training conditions. For instance, wav2vec 2.0 is commonly pre-trained with larger models on LibriSpeech to achieve lower WER on this dataset. This comparison can be found in appendix A.8 as it could bias the evaluation of our method with pre-training and target datasets being the same.

## 7 CONCLUSION

In this work, we introduce a method to quickly and simply combine pseudo-labels into a useful pretext task for multitask self-supervised learning settings. Our approach allows for an optimal selection of pseudo-labels following a cheap optimisation process drastically decreasing the time and compute needed to design the best performing multitask SSL model. Our method is validated on three speech-related downstream tasks and outperforms common pseudo-label selection strategies when combined with simple and state-of-the-art SSL models. This opens a range of possibilities for finding and selecting new pretext tasks in self-supervised learning for speech or other types of data.

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

## A  APPENDIX

### A.1  DIFFERENTIABILITY PROOF

We want to show that the utility estimate is differentiable with respect to the weighting parameters $(\lambda_i)_{i \in [0,k]}$. Since the final estimate is a weighted mean of the in-class independent tests, the problem boils down to showing that within a downstream class $c$, $HSIC_c(X, Z_\lambda)$ is differentiable. Let us recall the definition of the considered quantities:

$$HSIC_c(X, Z_\lambda) = \frac{1}{n_c^2} trace(K_c H_c L_c H_c) \tag{10}$$

where $K_c$ and $H_c$ are independent of $\lambda$ and $L_c$ coefficients are defined as:

$$[L_c]_{i,j} = RBF((Z_\lambda)_i, (Z_\lambda)_j) = \exp(\frac{-1}{2\sigma^2} \sum_{h=1}^{k} \lambda_h ||z_{h,i} - z_{h,j}||_2^2) \tag{11}$$

Therefore for $p \in [0, k]$ :

$$\begin{aligned}
\frac{\partial HSIC_c(X, Z_\lambda)}{\partial \lambda_p} &= \frac{1}{n_c^2} \sum_{i,j} \frac{\partial (trace(K_c H_c L_c H_c)}{\partial [L_c]_{i,j}} \frac{\partial [L_c]_{i,j}}{\partial \lambda_p} \\
&= \frac{1}{n_c^2} \sum_{i,j} (H_c^T K_c^T H_c^T)_{i,j} \frac{-||z_{p,i} - z_{p,j}||_2}{2\sigma^2} [L_c]_{i,j}
\end{aligned} \tag{12}$$

This allowed us to minimize the conditional-independence based utility estimator according to the weighting values.

Table 3: Candidate speech pseudo-labels and descriptions.

| Feature | Description |
| --- | --- |
| Loudness | Intensity & approx. loudness |
| F0 | Fundamental Frequency |
| Voicing | Voicing Decision |
| Alpha Ratio (Sundberg & Nordenberg, 2006) | Ratio of spectrum intensity % 1000 Hz |
| Zero Crossing Rate | Zero crossing number per frame |
| RastaSpec L1Norm | L1 Norm of Rasta Spectrum (Hermansky et al., 1992) |
| log HNR (Murphy & Akande, 2005) | log of Harmonicity to Noise Ratio |

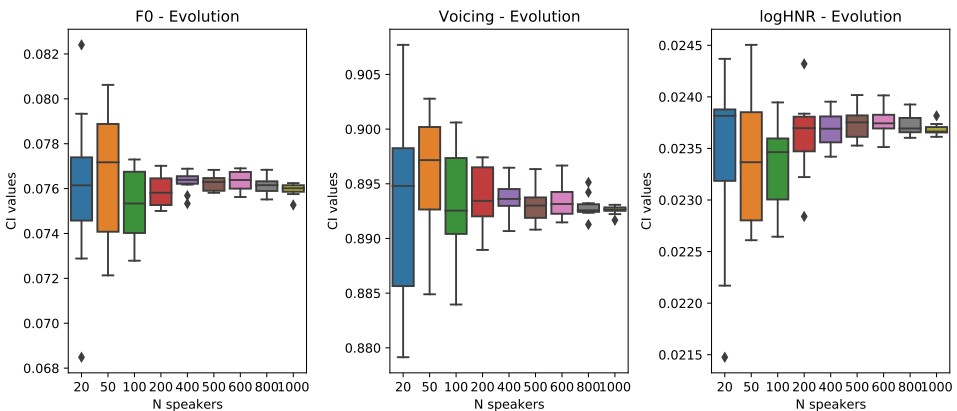

Figure 2: Evolution of the CI estimation with different numbers of considered speakers for three pretext tasks : F0, Voicing and logHNR. We can see that the values obtained with 20 speakers, while logically exhibiting more variance, are already close to the final values for every pretext task.

## A.2 CONSIDERED SIGNAL FEATURES AND DESCRIPTIONS

Table 3 contains the descriptions of the signal features used as pseudo-labels in this work.

## A.3 EFFECTS OF SAMPLING ON THE CONDITIONAL INDEPENDENCE ESTIMATION

Two limitations related to the size of the downstream dataset may be faced using our technique. First, very small downstream datasets could not be sufficient for a good estimate of the conditional independence. Second, very large downstream datasets may render the CI estimation intractable. This section shows experimentally on VoxCeleb1 that our technique is robust to these two situations. First, we show by taking small subsets of VoxCeleb1, that in case of downstream data scarcity, the CI estimations obtained with our method are close to the final estimations, and the ranking of the pretext tasks is not altered even when we take only 100 speakers among the 1 251 in VC. Second, as one of the main motivations of this work is the reduction of the computation needed to get the best selection of pretext-tasks in self-supervised learning settings, we show that the CI estimation converges quickly with a growing number of speakers considered, and is thus resilient to sampling. Considering one pretext task at a time, we consider subsets of VoxCeleb1 using a growing number of considered speakers ($total = 1\,251$). For each of these considered numbers, we run 10 experiments with sampled speakers. We get the CI estimation for every subset and plot the boxplot of the obtained values. Results are shown in Fig. 2. We can see that using only 20 speakers showcases results that are already close to those with 1 000 speakers. Furthermore, we plot the boxplots of CI values obtained using more than 200 speakers to show the separability between the considered features in Fig. 3. While values for Voicing and Loudness are slightly overlapping, all the other pretext tasks are already separated and rankable using only 200 random speakers among the whole dataset.

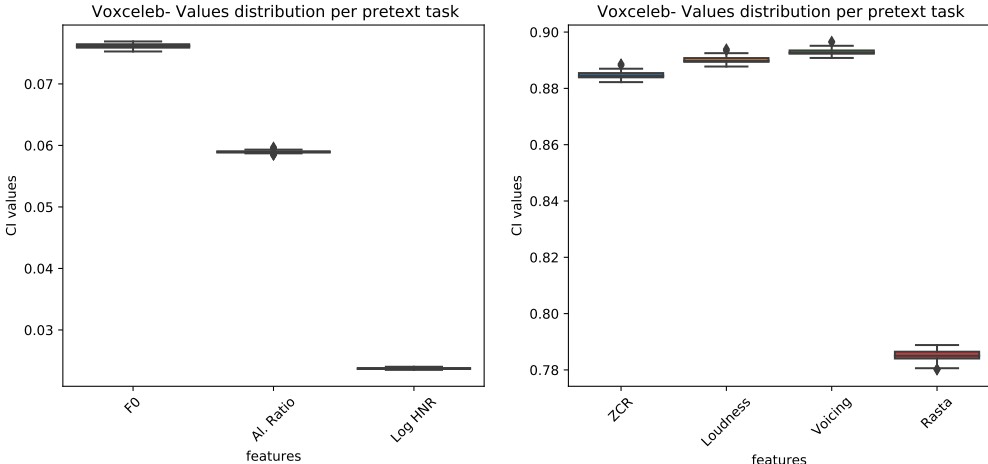

Figure 3: Boxplots of the CI values for every pretext tasks, when more than 200 speakers are considered. Voicing and Loudness are slightly overlapping, but otherwise, the values are separable. We divide the pretext-tasks in two groups according to their CI values for a better visualisation of the results.

## A.4 SPARSEMAX INITIALIZATION

When initialized with random parameters $W$, and if one parameter is high enough compared to the other, leading with the Sparsemax function to a weighting value close to 1, we observed that the minimization process falls into local minima selecting only one pseudo-label with weight 1. To avoid this, we initialize all the free parameters $W$ with the same unitary value to which we add some Gaussian noise. Hence, $W_{init} = (1) + N(0, \epsilon)$ with $\epsilon = 0.05$.

## A.5 TRAINING AND ARCHITECTURES DETAILS

All the considered audio files are sampled at 16kHz. We feed the SSL models with 80-band Mel spectrograms, with 25ms windows and 10ms stride. To every Mel band corresponds a learned vector of size 256 obtained at the output of the SSL model. So if the input spectrogram is of size $(N, 80)$ with $N$ the number of frames, the representation fed to the downstream pipeline is of size $(N, 256)$. All models including SSL and downstream ones are developed with SpeechBrain (Ravanelli et al., 2021).

### A.5.1 PRETRAINING OF THE SSL ENCODER.

The encoder is a succession of 2D CNN layers, LSTM layers and a final dense network. This representation is then fed to one dense layer that predict the selected pretext task labels. There are 3 successive CNN blocks containing each 2 CNN layers with kernel size $(3, 3)$ and 128, 200 and 256 channels for each block respectively. No time pooling is performed in order to preserve the input sequence length. 5 bidirectional LSTM layers of size 256 are then stacked. Finally, a MLP with one hidden layer with 256 neurons. The LeakyReLU activation is used across all the layers except for the LSTM. We use a dropout rate of $0.15$ during the training. The AdaDelta optimizer is used to update the weights with an initial learning rate of $1.0$, $\rho = 0.8$ and $\epsilon = 10^{-8}$. For every experiment, the SSL model is trained for 10 epochs ( leading to the convergence of the validation loss).

### A.5.2 DOWNSTREAM TRAININGS : FIRST EXPERIMENTS

**Speaker recognition details.** VoxCeleb1 (Nagrani et al., 2017) is used for the speaker recognition task. The training set contains $148, 642$ utterances from $1, 251$ different speakers. To compute the conditional independence estimates while limiting the computational load, we restricted ourselves to the utterances of 50 different speakers (the detailed list is given in the released repository). A standard xvector model (Snyder et al., 2018) is trained following the available VoxCeleb Speech-

Brain recipe. The extracted speaker embeddings are tested on the enrol and test splits using PLDA (Ioffe, 2006) as a similarity metric. Performance is reported in terms of equal error rate (EER). While architecture details are given in appendix A.5, it is worth noticing that the whole pipeline is fully integrated to Speechbrain and can thus easily be extended.

We train an embedding model (XVector) until the validation loss converges, on top of the self supervised representations using $5$ successive layers of time-delay neural networks (TDNN) (Peddinti et al., 2015). The number of channels is $(512, 512, 512, 512, 1500)$, with kernel sizes of $(5, 3, 3, 1, 1)$ and dilations of $(1, 2, 3, 1, 1)$. The architecture is inspired by successful works on embeddings for speaker recognition (Snyder et al., 2015). The learned embeddings are therefore used on a list of pairs of samples to predict whether they are from the same speaker or not. The details of the recipe can be found in the given GitHub repository. We train every embedding model on 10 epochs with an Adam Optimizer starting with a learning rate of 0.001 decaying linearly to 0.0001.

**Speech recognition details.** ASR is conducted with the 100-hour clean subset of the LibriSpeech dataset (Panayotov et al., 2015) to simulate the low-resource scenario commonly encountered with SSL settings. CI estimations are obtained with word-level alignments from the *Montreal Forced Aligner* (McAuliffe et al., 2017). The ASR pipeline follows the LibriSpeech recipe of SpeechBrain (Ravanelli et al., 2021) and therefore contains a CRDNN encoder (*i.e.* CNN, RNN, DNN) trained jointly with CTC (Graves, 2012) and attention (Lüscher et al., 2019) (details in appendix A.5). The decoding process is based on beam-search with and without shallow fusion with a pretrained recurrent language model that is publicly available and obtained from SpeechBrain.[2] Performance is expressed in word error rate (WER).

The CRDNN starts with three CNN blocks composed each with 2 2D CNN layers, layer-normalisation and $(2, 2)$ maxpooling along the frequency dimension. The filter dimensions for each block are $64, 100, 100$. Then, maxpooling of $4$ is applied on the time dimension to reduce the sequence length before being fed to the RNN. The latter is made of $5$ bidirectional LSTM layers of $1,024$ neurons. Finally two dense layers are connected (with batch-normalisation in between). The LeakyReLU activation function is used across all the layers except for the LSTM. A dropout rate of 0.15 is employed with the encoder. The CTC decoder is a simple dense linear layer of size equal to the vocabulary. The vocabulary is obtained with byte pair encoding or sub-words units (BPE) and is of size $1,000$. The attentional decoder is a one-layered location-aware GRU ($1,024$ neurons). Then, a beam search of depth $60$ is applied to obtain the output transcripts. The model is trained for $30$ epochs. The learning rate $(1.0)$ is multiplied with a factor of $0.8$ every time the validation loss is not decreasing to ensure an optimal convergence of all the models.

### A.5.3  SUPERB SETTINGS

SUPERB (Yang et al., 2021) is a recent benchmark for self-supervised representations of speech data. We use this benchmark for our experiments in combining wav2vec with our selected pretext tasks. We detail here the downstream models as detailed in the benchmark paper :

**Emotion Recognition.** IEMOCAP (Busso et al., 2008) is used for the Emotion Recognition (ER) task. 4 classes are considered (neutral, happy, sad, angry), and only the audio data is used. The learned representations are mean-pooled then fed to a final linear classifier to compute a cross-entropy loss. We cross-validate on five folds of the standard splits. The result shown is the average of the five attempts. The evaluation metric is accuracy (ACC).

**Automatic Speech Recognition** For ASR, the decoder is a vanilla 2-layer 1024-unit BLSTM fed with our self-supervised representations and optimized by CTC loss on characters. We use the same language model for decoding as in the first experiments. LibriSpeech Clean-100 only is used for downstream training.

**Speaker Recognition** The model and the dataset splits used in the first experiment correspond to the SUPERB ones, so we kept the same settings. The results are therefore comparable.

---

[2]https://huggingface.co/speechbrain/asr-crdnn-rnnlm-librispeech

## A.6 INTUITION AROUND THE USE OF CONDITIONAL INDEPENDENCE

To get an intuitive understanding of the motivations of this choice, let us consider the example of image classification as the downstream task, and image colourization as the pretext task. In this case, this pretext task would be suited to the downstream one if the final classification label can help implying the colours. For instance, if there are only two classes "Blue skies" and "Yellow deserts", then colourisation is an interesting pretext task, as knowing the final label helps a lot for the pretext task, independently of the image. However, if all the classes share the same colour palette, colourization may not be an interesting task. (In this simple example, we are ignoring the edge detection aspect of colourization, and only focusing on the colour choice part. Obviously the former aspect plays a big part on why the colourization pretext task has been successful.)

Concerning our estimation method, as the pseudo-labels considered in this work are data features, they are indeed functions of the original data samples. This ensures that the data samples are not independent of the pseudo-labels. The idea behind the estimator of conditional independence is that it will test whether this remains true when the considered points share the same downstream class.

## A.7 KERNELS USED FOR THE SIMILARITY MATRICES

The computation of the similarity matrices used in our kernel-based independence test, requires fixed-size embeddings for the data speech samples. These embeddings allow the use of classic kernels on top. However, in the case of sequential data, as it is the case with audio/speech signals, one may want to avoid the additional burden of learning fixed-size embeddings (for possibly variable-length audio sequences). One possible solution to this, which we conveniently exploited in our application to speech data (see Section 5) is the Gaussian Downsampling method (Holzenberger et al., 2018) detailed thereafter. In this instance, after the Mel spectrogram extraction, a speech sample is a sequence of varying length input feature vectors. Therefore, to obtain fixed size embeddings aggregating the input frame-wise Mel spectrum vectors into a fixed number $N$ of input vectors, $N$ being a fixed hyper-parameter, we first divide the sequence into $N$ equal length segments. Then, in each segment, a Gaussian average of the input spectra is computed around the center of the considered segment with the standard deviation $\sigma_{gd}$ being another hyper-parameter. Denoting by $D$ the dimension of the input frame-wise Mel spectrum vectors, this leads, for any speech excerpt, to a $N \times D$ tensor, without any training procedure. As in the work presenting the gaussian downsampling method (Holzenberger et al., 2018), we set $N = 20$ and $\sigma_{gd} = 0.07$. For the RBF kernel on the pseudo-labels mean value per file, we fixed he RBF kernel width to $\sigma = 0.05$.

We will know motivate the gaussian downsampling choice. At a high-level, we were looking for a method giving fixed size speech embeddings that would not rely on any learning. As we were first working with ASR and Speaker Id, we wanted a representation that captures phonetic and speaker content. The GD has been, quite successfully, applied for unsupervised word discovery (Holzenberger et al., 2018), thus capturing this phonetic aspect. In unpublished experiments when working on unsupervised work discovery, we also found out that GD holds speaker-related information, this made GD a good candidate for our unlearned embedding method. Moreover, after the first results, another downsampling method was tested in the beginning of this work : SVCCA (Raghu et al., 2017). We calculated the CI estimates using GD and SVCCA on the considered pretext tasks for LibriSpeech and Voxceleb1. It resulted in a relative difference of 3% between the averaged HSIC. The robustness to this downsampling method change comforted us into using GD.

## A.8 IMPACT OF THE PRETRAINING DATASET ON SPEAKER RECOGNITION

We developed in section 5.2 the reasons behind our choice of CommonVoice as a pre-training dataset. In this section, we study the impact of this choice on the results. Specifically, it is common in the speech SSL literature to train on LibriSpeech 960 before fine-tuning on LibriSpeech100. As explained before, we believe that this introduces a bias due to the closeness of pretraining and fine-tuning data. To verify this, we train our best multitask *BASE* wav2vec 2.0 architecture with the best performing pretext tasks and their weights on LibriSpeech 960. The model follows the exact same training procedure as for Table 2. We fine-tune the models on LibriSpeech 100 exactly as it has been done with the other models. Table 4 shows the results. Two observations deserve to be noted. First, in this case also, adding a selected set of pretext tasks improves the final downstream performance in the frozen and finetuned cases. Second, as expected, the results obtained after training on Lib-

Table 4: Results observed retraining the Wav2vec2 model with and without weighted pretext tasks using the sparsemax method, on LibriSpeech 960. "Fr." and "Fine." also respectively refer to Frozen and Finetuned settings. Adding selected pretext tasks still improves the downstream performance. All models contain $100M$ neural parameters.

| Selections | LibriSpeech *(WER % ↓)* | |
|---|---|---|
| | Fr. | Fine. |
| wav2vec 2.0 *BASE* | 9.88 | 6.33 |
| wav2vec 2.0 *BASE* + multitask SSL | **9.5** | **6.01** |

rispeech960 are better than those with CommonVoice, reaching the lowest $6.01\%$ of WER with the fine-tuned version compared to $9.18\%$ of WER in table 2.

## A.9 LINKS WITH FEATURE SELECTION

We also studied the link between classic feature selection and pretext task selection through two experiments. The first one was made to check how hard it was to estimate the utility of a pseudo-label, we computed the mutual information between the pseudo-labels and the downstream labels, and checked how much it would correlate with downstream performance. It led to very low correlation values, with even changing signs between VoxCeleb and LibriSpeech. This seems to indicate that Mutual Information is not related directly to self-supervision utility.

The maximum relevance minimum redundancy (MRMR) technique (Peng et al., 2005) used as a baseline in this work relies on the Conditional Independence based estimator. It is a close to a naive selection of the best pretext tasks according to the CI based criterion, but it furthermore penalizes the mutual information between the selected pretext tasks. More precisely, we select the group of pseudo-labels $(Z)_i \in [0, p]$ maximizing :

$$Score_{MRMR}(Z) = \frac{-1}{p} \sum_{i \in [0,p]} HSIC(X, Z_i | Y) - \frac{1}{\binom{p}{2}} \sum_{i<j} I(Z_i, Z_j)$$

Recursive Feature Elimination (RFE) (Guyon et al., 2002) relies on a classifier that provides information concerning the importance of a given feature in the decision. This classifier is first trained with the whole set of pseudo-labels as features, and the least important feature is eliminated. The process is repeated until only 4 pseudo-labels are kept. We use the scikit-learn implementation with the C-Support Vector Classification as the the classifier providing the feature importance values. We use the default scikit-learn hyperparameters.

These two baselines perform worse than the proposed techniques. This suggests that despite the apparent similarity, feature selection and self-supervision pretext task design do not necessarily involve the same mechanisms.

## A.10 PSEUDO-LABELS' INTERACTIONS.

To understand the interactions between pseudo-labels, studying the evolution of the CI estimate as a function of the weights shows which pseudo-labels seem interchangeable, which ones are complementary and which ones seem only harmful to the considered downstream task. Figure 4 shows the CI estimates for weighted combinations of groups of three pseudo-labels. As the weights sum up to one, two pretext tasks' values are shown on the $x$ and $y$ axes, while the value of the remaining one, whose name is in the title, is equal to $1 - x - y$. For instance, at the origin point $(0, 0)$, only the third pseudo-label is selected with a weight equal to one, while its weight is equal to zero on the hypotenuse of the right triangle. Figure 4 illustrates that the relationship leading to a lower CI-based utility estimator is not always straightforward. For instance, if we consider the second plot on the second row (*i.e. α-ratio, F0, logHNR*), we can see that selecting only one element is always worse

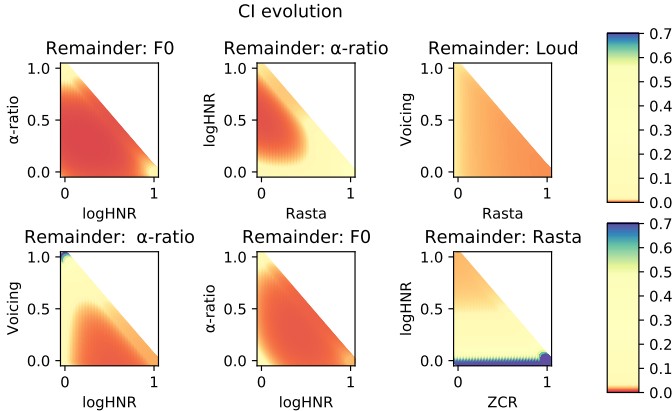

Figure 4: CI-Based utility estimator as a function of the weighting for groups of three pseudo-labels. Top line is for Librispeech, while the bottom one is for VoxCeleb. Three pseudo-labels are presented on every plot, one on the $x$-axis, one on the $y$-axis and one that is equal to $1 - x - y$ (hence being called the remainder) and whose name is on the title. Every point in the triangle corresponds to a pretext task that is the weighted combination of the three considered pseudo-labels. For instance, in the top left corner, the point $(0.5, 0.3)$ correspond to the CI value of a pretext task weighting logHNR with $0.5$, $\alpha$-ratio with $0.3$ and F0 with $0.2$.

Table 5: Weights for every pretext-tasks in every considered experiment. With techniques only leading to a selection of pretext tasks (without weights) a unitary weight is assigned for the selected tasks and zero for the non selected. We can see in this table the zeros induced by the Sparsemax function.

| Selection | $\alpha$-zero | F0 | Loudness | Audspec Rasta | ZCR | log HNR | Voicing |
|---|---|---|---|---|---|---|---|
| All | 1 | 1 | 1 | 1 | 1 | 1 | 1 |
| VC RFE | 1 | 1 | 0 | 0 | 1 | 0 | 1 |
| VC MRMR | 1 | 0 | 0 | 1 | 0 | 1 | 0 |
| VC Sparsemax | 0.28 | 0.26 | 0 | 0 | 0 | 0.4544 | 0 |
| VC Softmax | 0.27 | 0.11 | 0.18 | 0.04 | 0.06 | 0.31 | 0.03 |
| Libri RFE | 1 | 0 | 0 | 0 | 1 | 1 | 1 |
| Libri MRMR | 0 | 1 | 0 | 1 | 0 | 1 | 1 |
| Libri Sparsemax | 0.30 | 0.37 | 0 | 0.06 | 0 | 0.27 | 0 |
| Libri Softmax | 0.28 | 0.47 | 0.07 | 0.04 | 0.02 | 0.08 | 0.04 |
| IEMOCAP RFE | 0 | 0 | 1 | 1 | 1 | 1 | 0 |
| IEMOCAP MRMR | 0 | 1 | 0 | 0 | 1 | 1 | 1 |
| IEMOCAP Sparse. | 0.16 | 0.22 | 0 | 0.14 | 0.12 | 0.17 | 0.19 |
| IEMOCAP Softmax | 0.29 | 0.32 | 0.06 | 0.24 | 0.03 | 0.02 | 0.03 |

than selecting a weighted concatenation, because the areas around the origin and the points $(1, 0)$ and $(0, 1)$ are brighter than the central area.

A.11 FULL WEIGHTS FOR THE CONSIDERED EXPERIMENTS

Table 5 shows the weights computed using the proposed selection and weighting process for every considered downstream task. It also shows the pretext tasks selected by the baseline methods.

