# OpenReview forum: "Pretext Tasks Selection for Multitask Self-Supervised Speech Representation Learning"
_ICLR.cc/2022/Conference — ICLR 2022 Submitted_

### Official Review · Reviewer_zWkh · 2021-11-03

**Correctness:** 2
**Technical Novelty And Significance:** 3
**Empirical Novelty And Significance:** 2
**Recommendation:** 3
**Confidence:** 3

**Main Review:**

Strengths:
1) This topic is well timed. Currently, contrastive losses are the best performing, but having SSL targets from a multi-task scenario is also promising. It would be incredibly useful to the community if the weighting of many SSL targets was important, and one could be found to make multi-task training paradigms competitive.
2) The theoretical result seems interesting. I have not seen Conditional Independence be used as a criteria for weighting SSL targets. If useful, this would be a new way of thinking about tasks in a multi-task training scheme.

Weaknesses:
1) Empirical results:
1a) I worry about the comparison in Table 2. To support the claim that the authors’ algorithm is useful, they need to show that their algorithm also improves on wav2vec + naive mixing of multitask SSL. Table #2, as it currently stands, only shows that additional targets can be useful. It does not seem to support the argument that their mixing algorithm was essential to the improvement.
1b) I worry about the presence of IEMOCAP in experiment #2 but its absence in experiment #1. If the algorithm is as general as is claimed in the paper, I would expect feature selection improvements on IEMOCAP as well, and I would expect to see this dataset in table 1.

2) Unclear algorithm choices: The choice of the Gaussian Downsampling function is essential to computing the CI between X and Z given Y, but is unjustified in the text. Why take a Gaussian average with the hyper parameters used? These seem like essential ingredients to calculating the data component of conditional independence, but they are neither justified theoretically or experimentally (for instance, by determining how sensitive the final Z weights are to these parameters)

Additional nits:
1) There are some typos ex “( without weights )” -> “(without weights”
2) "The most successful models rely on predictive and contrastive objectives (Baevski et al., 2020; Chung et al., 2019; Saeed et al.,
2020)..." https://arxiv.org/abs/2110.04621 shows that "Saeed et al., 2020" is not better than other 2020 methods, and in many cases worse. It might be better to be more descriptive than "successful," which doesn't seem strictly true.

**Summary Of The Paper:**

This paper attempts to improve self-supervised (SSL) speech representations by determining what linear combination of SSL targets will perform best for downstream tasks (audio SSL targets like fundamental frequency or MFCC). The technique learns to weight these targets according to a Conditional Independence criteria: a target is more valuable if the SSL target is more independent of the data given a label.

There are two main experimental results on which the usefulness of this work rests. The first is a “sanity check”, which demonstrates that this weighting scheme outperforms two previous ones, from 2002 and 2005, on two tasks: ASR in LibriSpeech and speaker ID in Voxceleb1.  The second is that they manage to improve LibriSpeech, Voxceleb, and IEMOCAP performance over a wav2vec2 model by adding SSL targets weighted according to their algorithm.

**Summary Of The Review:**

The work is a very interesting idea in a relevant subdomain of SSL research. The idea of using Conditional Independence to rate SSL targets, and the technique for calculating CI, is novel. However, the empirical results do not support the usefulness of this method (experiment #2) or are not compared to strong baselines (experiment #1). Without stronger empirical results to support the usefulness or relevance of this method, I do not believe this paper is strong enough for acceptance.

---

> ### Author Response · Authors · 2021-11-23
> **Response to Reviewer 3 (Reviewer zWkh)**
>
> The authors thank Reviewer 3 for the efforts put into examining our work, and for judging that our work is well timed and that it can be incredibly useful. We will know address the concerns raised :
>
> >  Empirical results: 1a) I worry about the comparison in Table 2. To support the claim that the authors’ algorithm is useful, they need to show that their algorithm also improves on wav2vec + naive mixing of multitask SSL. Table #2, as it currently stands, only shows that additional targets can be useful. It does not seem to support the argument that their mixing algorithm was essential to the improvement. 1b) I worry about the presence of IEMOCAP in experiment #2 but its absence in experiment #1. If the algorithm is as general as is claimed in the paper, I would expect feature selection improvements on IEMOCAP as well, and I would expect to see this dataset in table 1.
>
> We understand the points raised by reviewer 3 and we ran additional experiments to answer their concerns. Regarding the first point, as judiciously suggested, we added a multitask + wav2vec experiment where the tasks are weighted like in the “all” setting, i.e mixed in a naive way ( it is called naive mixing in the new version ). Results are added to table 2. While the experiment shows that adding a naive selection already improves the performance of wav2vec2.0, it also highlights the benefit of  CI based weighting as it leads to the best performances.
> We also agree with reviewer 3 that the absence of IEMOCAP in the first table might seem suspect. It was mainly due to the computation costs involved by training 5 other self-supervised models. Nevertheless, and as requested, we ran these experiments. The results are now available in table 1 for IEMOCAP, and they confirm the results on the two other downstream tasks. We can also see that the results obtained are competitive with the baselines.
>
> >  Unclear algorithm choices: The choice of the Gaussian Downsampling function is essential to computing the CI between X and Z given Y, but is unjustified in the text. Why take a Gaussian average with the hyper parameters used? These seem like essential ingredients to calculating the data component of conditional independence, but they are neither justified theoretically or experimentally (for instance, by determining how sensitive the final Z weights are to these parameters)
>
> We fully understand this comment from reviewer 3. First, let us better explain and motivate this choice and then, highlight the changes made to the manuscript in that direction.  At a high-level, we were looking for a method giving fixed size speech embeddings that would not rely on any learning. As we were first working with ASR and Speaker Id, we wanted a representation that captures phonetic and speaker content. The GD has been, quite successfully, applied for unsupervised word discovery (https://hal.archives-ouvertes.fr/hal-01888708/file/Holzenberger_DKRD_2018_fixed_length_embeddings_for_words.Interspeech.pdf ), thus capturing this phonetic aspect. In unpublished experiments when working on unsupervised work discovery, we also found out that GD holds speaker-related information, this made GD a good candidate for our unlearned embedding method. Moreover, after the first results, another downsampling method was tested in the beginning of this work : SVCCA https://arxiv.org/abs/1706.05806. We calculated the CI estimates using GD and SVCCA on the considered pretext tasks for LibriSpeech and Voxceleb1. It resulted in a relative difference of 3% between the averaged HSIC. The robustness to this downsampling method change comforted us into using GD. We added a paragraph in appendix A.7 motivating this choice.
>
> > Additional nits:
> There are some typos ex “( without weights )” -> “(without weights”
> "The most successful models rely on predictive and contrastive objectives (Baevski et al., 2020; Chung et al., 2019; Saeed et al.,
>
> Thank you for spotting these typos, they have been corrected in the updated version.
>
> > ..." https://arxiv.org/abs/2110.04621 shows that "Saeed et al., 2020" is not better than other 2020 methods, and in many cases worse. It might be better to be more descriptive than "successful," which doesn't seem strictly true.
>
> We want to thank the reviewer for pointing out this impressive paper. Although we could not be aware of this work as it has been published after the ICLR submission deadline, we recognize we missed to cite a few baseline models cited there like TRILL and FRILL. We are adding these in the related works part in our new submission, as triplet-loss learning based methods. Apart from its results, we cited COLA as it is one of the first attempts to implement a contrastive loss close to what has been done in vision research with SimCLR. We added the citation of the Shor and al. work to the list of successful models, given the impressive results shown in the indicated paper.

---

### Official Review · Reviewer_ybV2 · 2021-11-06

**Correctness:** 3
**Technical Novelty And Significance:** 3
**Empirical Novelty And Significance:** 2
**Recommendation:** 6
**Confidence:** 3

**Main Review:**

This paper proposes a new approach to combine multiple tasks during pre-training for speech representation learning, and it leads to improved performance for downstream tasks including speech, speaker and emotion recognition. Overall, this paper is clearly written.

One question I think this paper needs to address is how strong baselines are. Though authors mention not to directly compare with downstream tasks' performance reported in literature, it still helps to provide some justification to show the numbers reported in Table 1 & 2 are improvements over strong baselines. Also authors should add error bar to show if difference is statistically significant.

**Summary Of The Paper:**

This paper describes authors' proposal for selecting pretext tasks (pseudo-labels) for multitask self-supervised learning, to improve self-supervised learning for speech representations. It leveraged the Hilbert Schmidt Independence Criterion (HSIC) for selecting pseudo-label loss weights and used softmax and its sparsity version to realize it. Experimental results on speech, speaker and emotion recognition showed effectiveness of the proposed approach.

**Summary Of The Review:**

The proposed method is technically sound, though additional justification is needed to for experimental results reported in this paper.

---

> ### Author Response · Authors · 2021-11-23
> **Response to Reviewer 2 (Reviewer ybV2)**
>
> The authors thank Reviewer 2 for considering that the paper is well written and recognizing the improvements that our work provides to the field. Positive  comments  are  very  much  appreciated!  Now,  we  would  like  to carefully  address  every  concern  raised  by  Reviewer 2  and  change  the manuscript accordingly.
>
> > One question I think this paper needs to address is how strong baselines are. Though authors mention not to directly compare with downstream tasks' performance reported in literature, it still helps to provide some justification to show the numbers reported in Table 1 & 2 are improvements over strong baselines.
>
> We added a few baseline results in Table 1 and Table 2 from the SUPERB self supervised learning benchmark that is the only standardized benchmark for SSL of speech representations in English. It is important to note that our experimental setup is more challenging and realistic than those reported as baselines as we are considering the CommonVoice dataset instead of LibriSpeech (this is demonstrated in [this work](https://arxiv.org/pdf/2104.14297.pdf)). Indeed, pre-training and fine-tuning on LibriSpeech would strongly bias our evaluation as shown in the added appendix A.8, so we moved the pre-training to another challenging and well-known dataset.
> > Also authors should add error bar to show if difference is statistically significant.
>
> Confidence intervals have been added on all the results to address this concern. In almost all the cases, the differences are statistically significant.

---

### Official Review · Reviewer_Q1kK · 2021-11-08

**Correctness:** 2
**Technical Novelty And Significance:** 2
**Empirical Novelty And Significance:** 2
**Recommendation:** 3
**Confidence:** 5

**Main Review:**

Strengths:
- the proposed method for selecting pretext tasks is interesting
- using this approach the authors show that self-supervised learning could benefit the fine-tuning for the downstream task

Weaknesses:
- evaluation for the claims are very weak. Authors show, for example, that using a subset of pretext tasks benefits the accuracy of ASR model. However, the experiments are limited to two downstream tasks. Also, results do not show the impact of picking other subsets which are not strongly related to the downstream task.
- could this approach be applied to other domains? or is it just limited to speech?
- comparison with wav2vec-2.0 is weak, can this approach beat the SOTA?
- are the learned features generable? or they fail when the downstream task is changed?
- how much data is required to learn the weights with HSIC?



**Summary Of The Paper:**

This paper proposes a method to select a group of pretext tasks from a given set of tasks for optimising the network training during self-supervised learning phase.
The weights for the given set of tasks are learned based on Hilbert Schmidt Independence Criterion (HSIC) using a few data samples. Using 2 downstream tasks, the authors show that their approach could benefit the learning of features relevant for the downstream tasks. Thus, improves the accuracy of these downstream tasks.

**Summary Of The Review:**

Overall, the approach is interesting and could benefit the training during self-supervised learning phase. However, the evaluation requires a significant amount of ablation studies to prove the claims about the effectiveness of this approach.

---

> ### Author Response · Authors · 2021-11-23
> **Response to reviewer 1 (Reviewer Q1kK) (1/2)**
>
> First,  the authors want to  thank  Reviewer  1  for  finding  our  method interesting, and for the questions raised that helped us improve the paper. In the following, we will carefully address each of her/his concerns   and  report  the  changes  made to  the manuscript:
>
> > evaluation for the claims are very weak. Authors show, for example, that using a subset of pretext tasks benefits the accuracy of ASR model. However, the experiments are limited to two downstream tasks. Also, results do not show the impact of picking other subsets which are not strongly related to the downstream task
>
> Answering reviewer 1  concern, and while experimentations are very computationally heavy ( every result requires a whole pretraining [from 4 to 20 GPUs] and fine-tuning [2 GPUs] ) we  added  new experiments validating our method. As proposed, we started by adding experiments on a third downstream task, emotion recognition, in table 1 (updated in the manuscript). Emotion recognition is another different task that relies on different aspects of the signal, and thus is a good complement to the other two downstream tasks chosen. As also suggested by reviewer 1, we tested a subset of tasks that are not related to the downstream task. These were chosen to be the ones which were deemed the least relevant by the MRMR subset selection. This has allowed us to see how much taking unrelated additional pretext tasks may harm the final downstream performance. As expected, and as can be seen in table 1, this subset was among the worst performing, hence validating empirically our approach.
>
> > could this approach be applied to other domains? or is it just limited to speech?
>
> We want to thank reviewer 1 for raising this point as we are also interested in this question as a long term goal. In short, yes, this approach can be applied to other types of data. For instance, in computer vision, multitask self supervision has been explored in previous works : (https://openaccess.thecvf.com/content_ICCV_2017/papers/Doersch_Multi-Task_Self-Supervised_Visual_ICCV_2017_paper.pdf) with pretext tasks  including colorization, jigsaw… These methods may benefit from a better weighting of the pretext tasks involved; this can be done using the technique developed in this paper. We went for speech data first, as it is our main field of interest, and second, because the literature on audio feature engineering offers a wide range of features whose prediction can be used as a pretext task, and whose utility has been proven in previous works like PASE+.
>
> > comparison with wav2vec-2.0 is weak, can this approach beat the SOTA?
>
> The experiments displayed in table 2 show that our method improves the results of wav2vec 2.0 which is the state-of-the-art for SSL of speech representations. Nevertheless, SOTA results rely on (very) large models trained on extremely large datasets, requiring computational infrastructure that most institutions do not even have access to. Still, we would like to highlight to reviewer 1 that a single of our experiments in the second table requires 24 GPUs for 2 entire days (NVIDIA tesla V100). One of the main advantages of  our method is actually that it avoids wasting huge amounts of compute (and time) by training the best model directly. Finally, we added an experiment where we changed the dataset used for pretraining from CommonVoice to Librispeech960. Librispeech is commonly used in speech SSL benchmarks for pretraining, before fine tuning on the clean 100-hours subset. It can be seen in the added appendix A.8 that it improves largely the results outperforming the wav2vec2.0 results published in the SUPERB benchmark.

---

> > ### Comment · Reviewer_Q1kK · 2021-11-26
> > **Thanks for the reply**
> >
> > One of the claims of the paper is that adapting the choices in the self supervised pipeline to a specific downstream task improves the final downstream performance. I agree with the claim that refining the pretext tasks improves accuracy for the downstream task. But my question about the impact of this refinement on other downstream tasks is still not answered. If we use these pretext tasks but train the model for another downstream task, what would be the impact? This could also tell us if your approach is finding certain pretext tasks that are not useful and shouldn’t be used.
> >
> > This statement is not clear: "it is not so relevant to test the generalisation of the models learned on data different from the downstream.” Why is the data different?
> >
> > I understand that training these models require a great number of gpu hours. In such a case, since you’re using a smaller dataset, why can’t we perform supervised training? Do you have numbers to show that even with smaller dataset your approach outperforms supervised learning based training?

---

> > > ### Author Response · Authors · 2021-11-27
> > > **Answer to Reviewer 1 (Reviewer Q1kK)**
> > >
> > > Thank you first for your reply. Here are precise answers to dissipate your final concerns.
> > >
> > > > Comment: One of the claims of the paper is that adapting the choices in the self supervised pipeline to a specific downstream task improves the final downstream performance. I agree with the claim that refining the pretext tasks improves accuracy for the downstream task. But my question about the impact of this refinement on other downstream tasks is still not answered. If we use these pretext tasks but train the model for another downstream task, what would be the impact? This could also tell us if your approach is finding certain pretext tasks that are not useful and shouldn’t be used.
> > >
> > > We understand your question and we added a small experiment to test the generalization to other downstream tasks in the first reply (see above). You may have missed it, which is fine as the answer was pretty developed, so here is a copy of what we added to the paper w.r.t this concern. **“To test even further our claim, we took the best model for librispeech (softmax weighting) and fine tuned it on Voxceleb1 and IEMOCAP for speaker and emotion recognition. While performing better than the simple baselines, it fails to reach the results obtained with the task-dependent weightings. We added this comment to section 5.4.”**  Here is a copy of the added paragraph to section 5.4 in paper :
> > > **To investigate how strongly improvements are correlated to the task, we took the best learned model for LibriSpeech (i.e. softmax weighting) and fine-tuned it on VoxCeleb1 and IEMOCAP. It reaches an EER of 10.55% and an accuracy of 59.9% respectively compared to 8.63 and 60.8 for the task-related experiments. While it performs better than the baselines, the difference between these results and the best performing selections shows that the weightings are indeed task-related.**
> > >
> > > > This statement is not clear: "it is not so relevant to test the generalisation of the models learned on data different from the downstream.” Why is the data different?
> > >
> > > You are right about this statement being unclear. In every model, the data we are learning on is the same. What we meant here is not the training data, but the downstream data used to compute the weights for the pre-training phase. There's probably still a misunderstanding here : the SSL training depends on the downstream labels, hence considering a different downstream task is against the logic of the method. Still, to answer your question and out of personal curiosity, we did the experiment explained in the previous paragraph and previous answer.
> > >
> > > > I understand that training these models require a great number of gpu hours. In such a case, since you’re using a smaller dataset, why can’t we perform supervised training? Do you have numbers to show that even with a smaller dataset your approach outperforms supervised learning based training?
> > >
> > > As demonstrated by the large literature on SSL for speech, self-supervised features outperform drastically acoustic features. This is also true in our experiments, and even with our small dataset, as we followed the well-known SUPERB benchmark that reports a WER of 23.2% on LibriSpeech with Mel filterbanks. The best models shown in our work and following our method reach 6.01% WER on Librispeech, 5.33% EER on Voxceleb1 and 74% accuracy on IEMOCAP. In comparison, training the downstream models with mel spectrograms leads to the following : 23.2% WER, 9.56% EER and 35.39% accuracy for emotion recognition. **It is important to see that our method can be plugged to the best performing model currently (wav2vec 2.0) and improves its performance, hence beating SOTA**.
> > > We would like to thank Reviewer 1 for the time taken during this discussion. It is great to finally engage with reviewers on this work. Please let us know if you any any further concerns. Otherwise, if all concerns have been addressed, we would like to encourage Reviewer 1 to increase the score to reflect all the changes that we did according to these solved issues.

---

> ### Author Response · Authors · 2021-11-23
> **Following the first response to Reviewer 1  (2/2)**
>
> > are the learned features generable? or they fail when the downstream task is changed?
>
> We understand the reviewer’s concern on this point. As a matter of fact, the majority of the speech embedding models published aim for universal speech representations. On the contrary, one of the particularities of this work is that every model learned with our technique is specific to the final downstream task to be solved. One of the claims of the paper is that adapting the choices in the self supervised pipeline to a specific downstream task improves the final downstream performance. The values weighting the losses during the pre-training phase are a function of the available downstream data. Thus, it is not so relevant to test the generalisation of the models learned on data different from the downstream. What can be seen however, is that the only model tested on every task, which is the model learning all the pretext tasks with constant weighting (called “all”) is one of the worst performing ones in every situation. To test even further our claim, we took the best model for librispeech (softmax weighting) and fine tuned it on Voxceleb1 and IEMOCAP for speaker and emotion recognition. While performing better than the simple baselines, it fails to reach the results obtained with the task-dependent weightings. We added this comment to section 5.4. Moreover, we can see that adding downstream-related tasks to the wav2vec2 loss outperforms the vanilla wav2vec2 and the naive wav2vec multi-tasked, this once again advocates for downstream task oriented choices in the self supervision part.
>
> > how much data is required to learn the weights with HSIC?
>
> Efficiency is one of the main motivations of this work. And thus, we share the concern of Reviewer 1 on two aspects, first the case where the downstream data is scarce, leading to very noisy CI estimation, second the case where the downstream data is too big, leading to intractable situations. To answer this concern, we added section A.3, with its two figures figure 2 and figure3, to the appendix with experiments sampling smaller and bigger parts from the Voxceleb1 dataset.  First, we show by taking small subsets of VoxCeleb1, that in the case of downstream data scarcity, the CI estimates obtained with our method are close to the final estimations, and the ranking of the pretext tasks is not altered even when we only take 100 speakers among the 1251 in VC. Second, as one of the main motivations of this work is the reduction of the computations needed to get the best selection of pretext-tasks in self-supervised learning settings, we show that the CI estimation converges quickly with a growing number of speakers considered, and is thus resilient to sampling. We hope this additional analysis answers the concerns raised by Reviewer 1.

---

### Author Response · Authors · 2021-11-23
**General response to the reviewers**

We would like to start by thanking all the reviewers for the efforts they have put into examining our work and spotting its strengths and weaknesses. We genuinely think that all their comments helped us significantly improve our submission by making the main ideas clearer and improving the justifications defending the proposed method and our claims.  **We took the time to systematically address each and every question or concern raised by all the reviewers, performing numerous costly additional experiments (hence the delay in the response), and we do hope that the updated version will dissipate the doubts concerning certain parts of the method and encourage the reviewers to update their scores accordingly if satisfied.**
Here is a list of the main additions to  the second version of the paper (every addition appears in red in the updated version):

* Added experiments on IEMOCAP in the first table confirming our results on ASR and speaker verification.
* Added experiments with naive mixing in the second table showing the benefit of a careful selection and weighting of the added pretext tasks.
* Added confidence intervals in all the results showing the significance of almost all the improvements.
* Added experiments with pretraining on LibriSpeech 960 closing the gap with SOTA results and showing our method improves the performance of wav2vec2.0 in the SUPERB benchmark.
* Added a study on the influence of the dowstream dataset size on the CI estimations. This study shows that our method is very robust to sampling and separates pretext tasks even in low-resource scenarios.
* Added experiments showing that our weightings are indeed downstream-tasks related by finetuning ASR best model for SV and ER and showing it performs worse than SV and ER best selection models.

More details about the additions/corrections and more additions are described in the individual answers provided to the reviewers.

---

### Decision · Program_Chairs · 2022-01-20

**Decision:**

Reject

**Comment:**

The paper proposes a method for selecting a group of pretext tasks out of a set of candidates in order to optimize self training for downstream performance. The method relies on Hilbert Schmidt Independence Criterion (HSIC) and uses a few data samples to select weights for the given set of tasks The paper demonstrates that using the method for task selection can result in learning better representation for downstream tasks improving accuracy on speech, speaker and emotion recognition.

The reviewers had concerns mostly about the strength of the empirical results. In particular, they felt that the baselines are not strong enough. To the authors credit, the paper was augmented with some of the missing experiments that the reviewers pointed out (e.g., wav2vec plus naive task selection), but that did not persuade reviewers to change their recommendations.

The paper still misses the point that self-supervised learning approaches can benefit from training larger models that result in better results. These comparisons are missing from the paper. It is established in other work that findings such as the use of pretext tasks often do not carry over to larger scales. Furthermore, the idea of pretraining a model specific to a downstream task is not inline of the philosophy of self-supervised training that aims to train a single model that can be used for many different tasks.